# COVID-19 and Cancer Detection in Russia

**DOI:** 10.3390/cancers16091673

**Published:** 2024-04-26

**Authors:** Andrey Sudarikov

**Affiliations:** National Medical Research Center for Hematology, Novy Zykovski Lane, 4a, 125167 Moscow, Russia; dusha@blood.ru; Tel.: +7-(926)-213-67-29

**Keywords:** COVID-19, cancer morbidity and mortality, screening, overdiagnosis

## Abstract

**Simple Summary:**

The COVID-19 pandemic caused a noticeable decrease in the number of cases of breast, prostate, renal, and thyroid cancers newly diagnosed in 2020 in the Russian Federation. There was no visible impact on mortality from these diseases. Mortality did not change during the pandemic and has remained at pre-pandemic levels until now. The detection rate returned to pre-pandemic levels during 2021–2022. One could speculate that the decrease in the number of newly diagnosed cases is due to cancer tests skipped during the lockdown. This is further evidence of the overdiagnosis of breast, prostate, renal, and thyroid cancers, which is linked to the mass testing of healthy populations.

**Abstract:**

Overdiagnosis, associated with mass testing in healthy populations, is a significant issue for breast, prostate, renal, and thyroid cancers. During the lockdowns caused by the COVID-19 pandemic, the intensity of cancer screening was expected to go down. In this study, we analyzed the impact of the expected reduction in screening intensity on morbidity and mortality from certain malignancies. Cumulative data from the Russian National Cancer Registry available from 2000 to 2022 were analyzed. It was noted that there has been no noticeable effect of the COVID-19 lockdowns on mortality rates from breast, prostate, renal, or thyroid cancers. At the same time, the detectable incidence decreased markedly in 2020 at the time of the lockdowns and then returned to pre-pandemic levels in 2022. At the moment, there is no sufficient reason to believe that skipping screening tests in 2020 could have any impact on breast, prostate, renal, or thyroid cancer mortality two years later (2022). The data presented further confirm that the overdiagnosis of these types of malignancies is caused by widespread screening among a generally healthy population.

## 1. Introduction

The issue of a balance between sufficient and excessive diagnostics in cancer has been the subject of discussion since the middle of the 20th century. In particular, George Grile’s claims that excessive concern leads to expensive research, inappropriate suffering, and unnecessary surgery have been opposed by many oncologists [1,2]. Great progress in the field of oncological screening in the second half of the 20th century is closely linked to the work of Sakary Timonen, who pioneered cervical cancer screening in Finland [3]. The use of the pap test, developed by Georgios Papanikolaou earlier [4], for screening healthy women has made it possible to reduce mortality from cervical cancer fivefold. The success sparked a wave of optimism. It seemed as if a way had been found to beat cancer—you just needed to find it in its early stages. Many new tests have been developed, and similar screening programs have been launched for other types of cancer, but the results have not always been promising.

The first experimental data on the existence of tumor cell reservoirs, which have little or no clinical significance, were obtained by Finnish pathologists [5]. Harach H.R. et al. showed that in 35.6% of thyroid autopsies of those who died from non-cancer causes, foci of “occult” papillary carcinoma could be detected. As it turned out, the situation with thyroid cancer is not unique and similar data have been reported for prostate cancer [6], breast cancer [7], and some others [8,9]. Due to the limitations of autopsy data, it is impossible to assess the dynamics of the detected tumor nodes. However, one can assume that these tumor nodes either develop slowly without reaching clinical significance for the patient’s lifespan or may not develop at all or even regress.

In addition to the autopsy studies, there is evidence for the overdiagnosis of certain types of malignancy based on the analysis of mortality and morbidity data. Gilbert Welch has demonstrated that, in regard to thyroid, breast, prostate, and kidney cancers, there has been a constant increase in the detection of these diseases; yet, mortality has not increased [10]. A more in-depth analysis of these data clearly shows that we are dealing with an overdiagnosis of certain diseases, rather than an improvement in the results of treatment. In particular, it has been shown that the initial detection of small tumor nodules in the screened cohort did not lead to a reduction in the number of significant findings at subsequent stages of screening. Screening mainly detects small, slowly growing tumor nodes. However, rapidly growing ones, which are of the greatest clinical significance, manage to develop between tests and are therefore beyond the reach of screening. In addition, the temporary increase in morbidity in certain malignancies coincides with the introduction of new diagnostic techniques.

Irrefutable evidence for the overdiagnosis of certain malignancies is also confirmed by long-term observations on the impact of cancer screening in different countries [11,12]. It turned out that in different countries with similar levels of medical care, the mortality rate from breast cancer did not depend on the time when screening tests were introduced, which varied by decades. The publications mentioned above describe observational data that, in principle, allow for an alternative interpretation. Interventional, randomized trials have also been carried out and a review of seven trials involving 600,000 women between the ages of 39 and 74 has been published by Peter Gøtzsche [13]. It was estimated that out of 2000 women invited for screening over 10 years, only one will avoid death from breast cancer. Additionally, 10 healthy women who would not have been diagnosed if there had been no screening will be undergo unnecessarily treatments including surgery, radiation therapy, chemotherapy, or a combination. Moreover, the lifestyle and mental health of more than 200 women will be dramatically disturbed by false diagnoses, disturbing expectations, and repetitive tests. Actually, one-half of the women may receive at least one “false alarm” and about half of those will undergo a biopsy [14].

Another case of excessive diagnosis of oncological diseases was caused by the widespread introduction of thyroid cancer screening. In particular, in South Korea, the incidence of thyroid cancer has increased by an average of 15-fold in the country. At the same time, mortality has remained unchanged. Moreover, the increased incidence in different regions was correlated with the penetrationof screening [15]. A similar correlation can be observed for kidney cancer diagnosis in the United States. The more frequently abdominal computed tomography (CT) scans are performed, the greater the likelihood of nephrectomy [16]. For every thousand men who undergo regular screening for prostate cancer, one death can be prevented. Four men will die from prostate cancer despite treatment, while 53 others will receive treatment but would have survived without it. Thirty-four men will be treated but their cancer will not manifest symptoms during their lifetime. One hundred and fifty men will undergo the stress of a false diagnosis and additional tests, such as a biopsy [17,18]. It should be noted that screening might be associated with overdiagnosis only for certain types of malignancies such as thyroid, prostate, breast, and kidney cancers. The value of screening tests for early diagnosis and the prevention of other oncological diseases is undeniable. For example, a 30-year follow-up of a cohort of patients showed a significant decrease in colorectal cancer mortality in the group of patients who received a screening test compared to those who were not screened. Moreover, a dose-dependent effect was demonstrated: annual testing provides more benefits than biennial testing [19].

The COVID-19 pandemic had a significant impact on public health. This affected both those who were directly affected by the infection and healthy people who postponed their visits to clinics due to quarantine measures. Doctors were also distracted from performing routine tests and procedures. Instead, many clinics and specialists were engaged in fighting the infection. As a result, access to preventive screening for oncological diseases was limited. We attempted to assess the extent to which these restrictions affected cancer morbidity and mortality based on statistical data for the Russian population.

## 2. Materials and Methods

We collected data on breast, prostate, renal, and thyroid cancer incidence and mortality from the Russian National Cancer Registry from 2000 to 2022 [20]. At the time of writing, the 2022 report on morbidity and mortality was finalized and published [21]. The absolute number of new cases diagnosed each year, along with the reported deaths, were plotted on charts using LibreOffice software (version 7.3.7.2) [22].

## 3. Results

Absolute, unadjusted data on cancer incidence and mortality from 2000 to 2022 were used to create charts. The numerical data are provided in the Appendix A. Plots for the total number of breast, prostate, renal, and thyroid cancer cases are shown in Figure 1. The data presented indicate that in 2020, there was a significant decrease in the number of newly reported cases for the mentioned malignancies. At the same time, there was no significant effect on mortality from these diseases, either immediate or delayed. In 2021, the number of new cases began to increase and by 2022, it had almost reached pre-pandemic levels. On the other hand, the mortality rate from these diseases remained unchanged throughout this entire period of time. It should be noted that due to the delayed association between morbidity and mortality, a more accurate assessment of the impact of skipping screening tests should be conducted after a longer period of follow-up. However, we believe that the trend observed so far (the lack of any significant impact) is worthy of the attention of researchers. Similar trends for breast, prostate, renal, and thyroid cancers can be observed using other datasets, such as those provided by the SEER [23].

## 4. Discussion

Overdiagnosis is a significant issue in the context of mass screening for breast, prostate, kidney, and thyroid cancer. This is due to the presence of reservoirs of tumor cells, which are of negligible clinical significance but can be detected in healthy individuals using sufficiently sensitive methods. Neoplasms “diagnosed” in this manner develop slowly and are unlikely to cause significant health problems during a person’s lifetime. It should be noted that rapidly growing lesions may elude periodic testing and become clinically significant between tests. Conversely, positive cases identified during screening may include those that grow more slowly and have less clinical significance [10]. The initiation of aggressive treatment for these indolent conditions could negatively affect quality of life and also have a negative impact on economic efficiency in healthcare. As treatment methods improve and new targeted drugs become available, the risk of harm to patients due to overdiagnosis decreases. However, the negative impact of over-screening on healthcare costs will only increase as more modern and expensive treatments become available.

It should be noted that both among the general public and in the professional medical community, there is a prevalent misconception about the unconditional usefulness of early cancer diagnosis. Of course, the early detection of cancer increases the effectiveness of therapy and the chances of recovery in general. However, it is important to remember that there are still limitations to this approach [24]. One has to consider the potential risks of overtreatment and the possible side effects of anticancer therapy, along with the benefits of early cancer detection. Furthermore, the significance of early cancer detection cannot be assessed by survival time statistics. First, as a screening test detects a case before the onset of clinical symptoms, the time from the screening to the onset of symptoms will be automatically added to the duration of the illness. This is known as the “lead time bias” [25]. On the other hand, screening tests are effective at detecting slow-growing neoplasms of low (or no) potential danger while skipping aggressive, fast-growing tumors. Thus, thanks to screening tests, tumors that are not significant will mainly be “cured”. This is another bias in assessing the effectiveness of screening in a healthy population. Therefore, the reduction in mortality as a result of screening implementation, rather than prolonged survival, could be a strong argument in favor of mass screening programs [12]. Unfortunately, in some cases, we do not see a decrease in mortality as a result of the introduction of screening. For example, in Sweden, mammography was introduced almost 20 years earlier than in Norway (a country with a similar level of medical technology), but no difference in breast cancer mortality was found over the follow-up period [11].

COVID-19 has had a significant negative impact on cancer treatment. Patients who were treated during the pandemic were more likely to become infected due to being immunocompromised and the need to attend hospitals and clinics. However, not all cancer patients were affected equally. In particular, for patients with hematological malignancies, the risk of a poor outcome from infection was twice that of other cancer patients [26]. On the other hand, the COVID-19 pandemic has had a significant impact on public health in terms of access to preventive screenings for cancer. Skipping screening tests resulted in a significant decrease in the number of breast, prostate, kidney, and thyroid cancer cases newly diagnosed in 2020 in the Russian Federation. However, there has been no noticeable effect on mortality rates for these diseases. In fact, mortality during the pandemic and after remains at the same levels as before the pandemic. The detection rate returned to pre-pandemic levels in 2021 and 2022.

The decline in newly diagnosed cases is obviously due to the decreased intensity of cancer screening during the lockdowns. There are concerns that a decrease in cancer screening due to the pandemic could lead to an increase in cancer-related deaths in the future [27]. However, at this point, such fears appear to be exaggerated and have little basis, at least with regard to breast, prostate, kidney, and thyroid cancers. It is worth noting that even in a simulation study, only a small number of additional breast cancer deaths among women in the United States were predicted between 2020 and 2030 due to disruptions in the screening, diagnosis, and treatment of breast cancer caused by the COVID-19 pandemic [28]. The statistical data presented here further confirm that overdiagnosis in these types of malignancies is caused by widespread screening among a generally healthy population.

Despite the fact that many professionals are aware of the potential risks associated with overdiagnosis and overtreatment, it may sometimes be safer to prescribe unnecessary therapy rather than recommend monitoring without intervention due to legal concerns. Possible financial incentives, as well as the undeniable public belief in the importance of early diagnosis and treatment, are also significant factors [16,29]. The stories of “survivors”, which are replicated in the media and whose screenings “saved their lives” also play a significant role in the over-promotion of mass screening. For those who are involved in making decisions about healthcare, these stories are sometimes more convincing than statistical reports, scientific articles, or expert opinions. It is important to emphasize once again that in each case of screening, we cannot accurately determine whether we are dealing with a potentially dangerous tumor or with overdiagnosis. Therefore, we can only rely on population data, and the prognosis for individual patients is probabilistic. Even if the likelihood of harm exceeds the likelihood of benefit, in each case, the decision to undergo screening should be made jointly with the patient on the basis of information about potential consequences. Some patients decide to undergo screening even if they have complete information. The decision ultimately depends on personal values because, for many people, prolonging life at any cost is more important than the potential quality of life [30]. Doctors should respect the patient’s choice, whatever it may be. At the same time, it is important to resolutely counteract the propaganda of mass screening, which has an ambiguous balance of benefits and risks when they are advertised by medical centers or manufacturers of diagnostic equipment as an unconditionally useful diagnostic procedure.

## 5. Conclusions

The negative impact of the COVID-19 pandemic on access to cancer prevention screenings may be overestimated for some cancers. Overdiagnosis associated with the mass testing of healthy populations remains a significant concern for a number of malignant diseases. Data from the Russian National Cancer Registry support this point, as reported here.

## Figures and Tables

**Figure 1 cancers-16-01673-f001:**
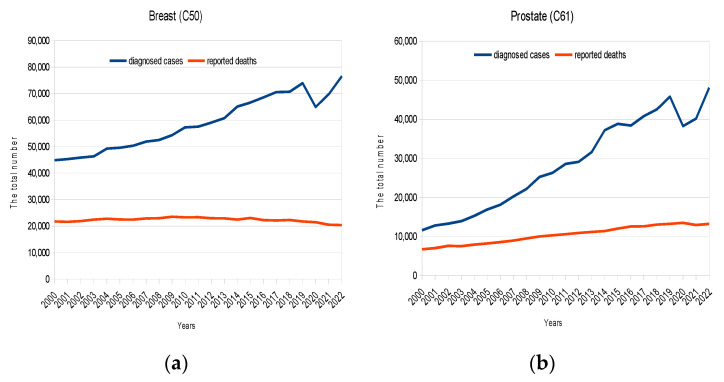
Cancer incidence and mortality in Russia from 2000 to 2022: (**a**) breast; (**b**) prostate; (**c**) renal; (**d**) thyroid. The total numbers of newly diagnosed cases are plotted in blue; reported deaths are shown in orange.

## Data Availability

Statistical data are available from the Russian National Cancer Registry [20].

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
