# Peer review of "COVID-19 and Cancer Detection in Russia"

_cancers, 2024, doi:10.3390/cancers16091673_

Round 1
Reviewer 1 Report
Comments and Suggestions for Authors
This a fascinating and very important topic. Screening for prostate cancer is especially difficult. The author points out that some individuals have fast moving cancers that can metastasise between screenings. This can occur if the approach is: "The PSA is up a bit lets recheck in 6 months" but some unfortunate individuals may have metastases identified on retesting 6 or 12 months later. "
The key and most interesting part of this article is this section. I would appreciate it if the author could critically review in detail the study designs and risks of bias of these important studies that he has commented on very briefly and uncritically. We count on our authors to identify those RCTs and other study designs at minimal risk of bias and that were carefully planned, executed and analysed.:
The authors wrote: "Harach HR et al. have shown that in 35.6% of 42 thyroid autopsies of those who died from non-cancer causes, foci of "occult" papillary 43 carcinoma can be detected. As it turned out, the situation with thyroid cancer is not 44 unique and similar data have been reported for prostate cancer [6], breast cancer [7] and 45 some others [8, 9]. Due to the limitations of autopsy data, it is impossible to assess the 46 dynamics of the detected tumor nodes. However, one can assume that these tumor 47 nodes either develop slowly without reaching clinical significance for the patient's life 48 Citation: To be added by editorial staff during production. Academic span, or may not develop at all if not even regress. In addition to the autopsy studies, 49 there is evidence for overdiagnosis in certain types of malignancy based on the analysis 50 of mortality and morbidity data. Gilbert Welch has demonstrated that, in regard to 51 thyroid, breast, prostate, and kidney cancers, there has been a constant increase in the 52 detection of these diseases, yet mortality has not increased [10]. A more in-depth analysis 53 of these data clearly shows that we are dealing with an overdiagnosis of certain diseases, 54 rather than an improvement in the results of treatment. In particular, it has been shown 55 that the initial detection of small tumor nodules in the screened cohort did not lead to a 56 reduction in the number of significant findings at subsequent stages of screening. 57 Screening mainly detects small, slowly growing tumor nodes. However, the rapidly 58 growing ones, which are of the greatest clinical significance, manage to develop between 59 tests and are therefore beyond the reach of screening. In addition, the temporary 60 increase in morbidity in sertain (Spelling!) malignancies coincides with the introduction of new 61 diagnostic techniques. Irrefutable evidence for overdiagnosis in certain malignancies is 62 also confirmed by long-term observations on the impact of cancer screening in different 63 countries [11, 12]. It turned out that, in different countries with similar levels of medical 64 care, the mortality rate from breast cancer did not depend on the time when screening 65 tests were introduced, which varied by decades. The publications mentioned above 66 describe observational data that, in principle, allow an alternative interpretation. 67 Interventional, randomized trials have also been carried out and a review of seven trials 68 involving 600000 women between the ages of 39 and 74 has been published by Peter 69 Gøtzsche [13]. It was estimated that out of 2000 women invited for screening over 10 70 years, only one will avoid death from breast cancer. Meanwhile, 10 healthy women who 71 would not have been diagnosed if there had been no screening will be treated 72 unnecessarily. Moreover, the health of more than 200 women will be significantly 73 harmed by unnecessary surgery, radiation, chemotherapy, or a combination. Actually 74 one half of the women may receive at least one "false alarm", and about half of those will 75 undergo a biopsy [14]. Another case of excessive diagnosis in oncological diseases was 76 caused by the widespread introduction of thyroid cancer screening. In particular, in 77 South Korea, the incidence of thyroid cancer has increased by an average of 15-fold in 78 the country. At the same time, mortality has remained unchanged. Moreover, the 79 increased incidence in different regions was correlated with the penetration of screening 80 [15]. A similar correlation can be observed for kidney cancer diagnosis in the United 81 States. The more frequently abdominal computed tomography (CT) scans are 82 performed, the greater the likelihood of nephrectomy [16]. For every thousand men who 83 undergo regular screening for prostate cancer, one death can be prevented. Four men 84 will die from prostate cancer despite treatment, while 53 others will receive treatment 85 but would have survived without it. Thirty-four men will be treated but their cancer will 86 not manifest symptoms during their lifetime. One hundred and fifty man will undergo 87 the stress of a false diagnosis and additional tests, such as a biopsy [17,18]. It should be 88 noted that screening might be associated with overdiagnosis only for certain types of 89 malignancies such as thyroid, prostate, breast, and kidney cancers. The value of 90 screening tests for early diagnosis and prevention of other oncological diseases is 91 undeniable. For example, 30-year follow-up of a cohort of patients shows a significant 92 decrease of colorectal cancer mortality in the group of patients who received a screening 93 test compared to those who were not screened. Moreover, a dose-dependent effect has 94 been demonstrated: annual testing provides more benefits than biennial [19]."
Reviewer 2 Report
Comments and Suggestions for Authors
Thank you for your submission. I hope you find the following comments constructive and valuable.
ABSTRACT: I would quantify your findings. For example, there is no sufficient reason to believe that skipping screening tests in 2020 could have any impact on mortality two years out (2022).
INTRODUCTION: The content is good but hard to read. From a stylistic standpoint, break the large single paragraph into multiple smaller paragraphs, each discussing one of your primary points. Overall, however, it is excellent.
METHODS: Please clarify if there is a delay in reporting mortality statistics. Was the 2022 registry finalized at the time of your data collection?
RESULTS: Consider including a table rather than just the graphs in the results. For example, have a table with columns showing the incident rates of breast CA, prostate CA, renal CA, and thyroid CA by year.
Do you have any more results? Have any statistical analyses been done?
DISCUSSION: typically the first paragraph of the discussion will give the study's primary results. Again, break down your ideas into separate paragraphs instead of putting everything into one long paragraph. Also, please include a separate paragraph stating the limitations of your research. Specifically, be sure to address the issue of time delay from screening diagnosis to mortality.
Thank you again for submitting your work. I encourage you to continue pursuing research in this critical area of cancer epidemiology and public health.
very long paragraphs in intro & discussion are hard to read
Round 2
Reviewer 1 Report
Comments and Suggestions for Authors
Hello Andrey,
Its fine that you know the articles have been critically assessed. But there are many readers around the world who have no access to university libraries. We also need your personal assessment of the risk of bias IN DETAIL. Its not enough just to quote that risk of bias has been assessed. We are no further ahead than in your original article. Now for the hard work of your assessment of the risk of bias and comparison and summarising what other assessor said.
Reviewer 2 Report
Comments and Suggestions for Authors
I appreciate your updates and agree that although a 2 year follow-up is early, it is still important information.
Thank you for your hard work and I look forward to your longer term follow-up of this important issue.
Author Response
Dear Sir or Madam,
Thank you for your positive feedback on our work and your support for our future research endeavors.
Yours faithfully, Andrey Sudarikov
Round 3
Reviewer 1 Report
Comments and Suggestions for Authors Dear Author, You have an interesting idea here. However, in your reply to my criticisms you just said that all these studies are high quality and in well known journals. This is not satisfactory: we expect you to analyse the evidence. For the randomised controlled trials use the Cochrane risk of bias tool on the Cochrane website to assess the study design, randomisation, concealment, blinding of participants, intervention administrators and outcome assessors and attrition, and for study quality assess with the GRADE tool on the GRADE website. For non randomised designs use the Newcastle-Ottawa scale. Thanks
